# Puzzles: Unbounded Video-Depth Augmentation for Scalable End-to-End 3D Reconstruction

**Jiahao Ma**[1,3]**, Lei Wang**[2,3]**, Miaomiao Liu**[1]**, David Ahmedt-Aristizabal**[3]**, Chuong Nguyen**[1,3]
[1]Australian National University, [2]Griffith University, [3]Data61/CSIRO
{jiahao.ma, miaomiao.liu}@anu.edu.au, l.wang4@griffith.edu.au
{david.ahmedtaristizabal, chuong.nguyen}@data61.csiro.au

## Abstract

Multi-view 3D reconstruction remains a core challenge in computer vision. Recent methods, such as DUSt3R and its successors, directly regress pointmaps from image pairs without relying on known scene geometry or camera parameters. However, the performance of these models is constrained by the diversity and scale of available training data. In this work, we introduce **Puzzles**, a data augmentation strategy that synthesizes an unbounded volume of high-quality, posed video-depth data from just a single image or video clip. By simulating diverse camera trajectories and realistic scene geometry through targeted image transformations, Puzzles enhances data variety. Extensive experiments show that integrating Puzzles into existing video-based 3D reconstruction pipelines consistently boosts performance, all without modifying the underlying network architecture. Notably, models trained on only **10**% of the original data, augmented with Puzzles, still achieve accuracy comparable to those trained on the full dataset.[**Project website**] [**Code**]

## 1 Introduction

Dense 3D reconstruction is a fundamental problem in computer vision, aimed at recovering detailed scene geometry from images or videos. Traditional approaches, based on Simultaneous Localization and Mapping (SLAM) [1–3], Structure-from-Motion (SfM) [4], and Multi-View Stereo (MVS) [5–7], typically follow a multi-stage pipeline involving camera pose estimation, correspondence matching, and dense depth inference. While effective in controlled environments, these methods often struggle with scalability, computational efficiency, and robustness to viewpoint changes, limiting their applicability in large-scale, real-world scenarios.

Recent advances in learning-based two-view geometry address some limitations of monocular dense SLAM. DUSt3R [8], for instance, introduced an end-to-end pipeline that directly regresses pointmaps from image pairs. However, it requires costly global alignment for multi-view reconstruction. Follow-up methods have improved on efficiency and scalability: Spann3R [9] replaces optimization-based alignment with incremental pointmap fusion in a shared coordinate frame using spatial memory; SLAM3R [10] reconstructs local geometry within a sliding window and integrates it incrementally into a global map; Fast3R [11] scales further, processing thousands of frames per forward pass without the need for alignment. Despite their advances, the *3R*-series methods depend heavily on large volumes of high-quality posed video clips with ground-truth depth. Unfortunately, such datasets [12–14] remain limited in diversity and scale, ultimately constraining model generalization.

To address these challenges, we introduce **Puzzles**, a data augmentation framework for 3D reconstruction that increases data diversity while reducing dependency on large-scale video datasets. Rather than relying on redundant video frames (Figure 1.A), Puzzles extracts and amplifies the geometric and photometric cues already presented in a single image (Figure 1.B.1). Inspired by the making of jigsaw-puzzle, Puzzles partitions an image into an *ordered*, *overlapping* sequence of patches. The

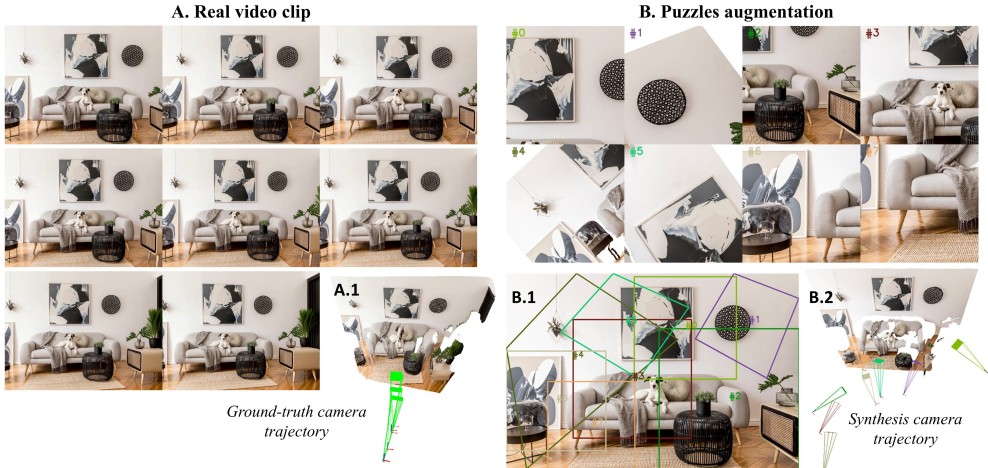

Figure 1: **Puzzles**: A new data augmentation framework for training feedforward 3D reconstruction deep models. **A**. Existing video-based 3D reconstruction datasets often contain redundant frames due to fixed camera trajectories after recording (**A.1**). **B**. Puzzles increases view diversity by generating novel viewpoints from a single image (**B.1**) and simulates realistic camera trajectories (**B.2**) to support 3D reconstruction training.

ordering encodes pseudo-temporal structure, while overlapping regions ensures spatial consistency, together simulating the geometric characteristics of real video sequences. As illustrated in Figure 1.B.2, these patches are further augmented to simulate realistic camera rotations and translations, producing diverse, video-like clips with synthetic depth and pose. We refer to this process as *Image-to-Clips*.

While Image-to-Clips enables fine-grained local reconstruction, a single image does not provide sufficient spatial coverage for large-scale scenes. To overcome this limitation, we propose *Clips-to-Clips*, an extension that operates on video clips by selecting strategic keyframes to generate multiple diverse sub-clips. This balances spatial coverage with augmentation diversity and mitigates the continuity issues caused by random frame sampling in prior works [9, 10].

Extensive experiments show that integrating Puzzles into existing video-based *3R*-series pipelines consistently improves reconstruction quality under the same training data budget. Remarkably, models trained on just **10**% of the dataset, when augmented with Puzzles, match or exceed the performance of models trained on the full dataset. Notably, Puzzles requires no changes to network architectures, making it a drop-in, plug-and-play solution. Our **contributions** are:

   i. We propose **Puzzles**, comprising *Image-to-Clips* and *Clips-to-Clips*, which generates realistic and diverse video-like clips with estimated depth and pose from single images or video clips via ordered, overlapping patch-based augmentation.

  ii. We show that integrating **Puzzles** into video-based *3R*-series pipelines consistently improves performance, even under reduced data budget, with models trained on just **10**% of the data achieving comparable or better results than full-data baselines.

 iii. **Puzzles** is a plug-and-play augmentation method that can be applied to any learning-based dense 3D reconstruction pipeline, without modifications to the underlying architecture.

## 2   Related Works

**End-to-end dense 3D reconstruction.** DUSt3R [8] presents the first fully end-to-end framework for dense 3D reconstruction without requiring explicit camera calibration. Building on this, subsequent works have adapted the same paradigm to tasks, *e.g.*, single-view reconstruction [15], feature matching [16], novel-view synthesis [17–19], camera pose estimation [20], and dynamic scene reconstruction [21–25], highlighting the general utility of dense point prediction. However, extending DUSt3R to video reconstruction remains both computationally expensive and error-prone. To address this, Spann3R [9] processes frames sequentially using a sliding window and external memory, while

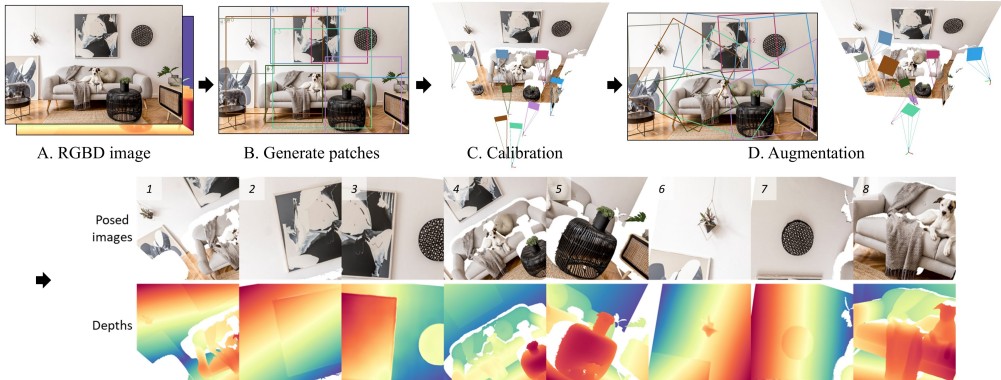

Figure 2: **Puzzles: Image-to-Clips.** (**A**) Starting from a single RGB-D image, we (**B**) partition it into ordered, overlapping patches, (**C**) simulate diverse viewpoints by calibrating virtual camera poses, and (**D**) generate augmented, posed images with aligned depth maps for use in 3D reconstruction.

Slam3R [10] incrementally registers overlapping clips. Fast3R [11] further improves scalability by reconstructing hundreds of unordered views in a single forward pass. Despite architectural advances, these *3R*-series works [8, 16, 26, 9–11, 27, 28] require vast amounts of posed video depth data or image pairs extracted from resources that we found still limited in both diversity and quantity. Unlike prior work focused solely on model design, we adopt a data-centric strategy: augmenting real data to synthesize rich, varied training samples that substantially improve model performance.

**Data augmentation for deep learning.** Data augmentation is essential in deep learning, particularly when labeled data is scarce. Accordingly, it has been widely adopted across numerous computer vision tasks. In image classification, methods such as image erasing [29–31] typically remove sub-regions to improve invariance, while image mixing techniques [32–35] blend multiple images or patches into one. For object detection, region-level transformations coupled with bounding-box adjustments [36] improve localization accuracy. In segmentation, ClassMix [37] synthesizes new examples by exchanging class-specific regions across unlabeled images. For visual tracking, MASA [38] and BIV [39] simulate pseudo frame pairs by augmenting static images to emulate motion. Despite these advances, data augmentation for dense 3D reconstruction remains underexplored. To address this gap, we introduce Puzzles, a method that synthesizes full video sequences with corresponding camera poses and depth maps from a single image or short clip. Unlike prior methods [38–40] that focus on generating frame pairs, Puzzles create temporally coherent video streams that offer rich 2D viewpoint variation while preserving geometric consistency in 3D.

## 3 Method

Given a monocular video consisting of a sequence of RGB frames $\left\{I^n \in \mathbb{R}^{H \times W \times 3}\right\}_{n=1}^N$ capturing a static scene, the goal is to reconstruct a dense pointmap $\left\{X^n \in \mathbb{R}^{H \times W \times 3}\right\}_{n=1}^N$, where each $X^n$ is represented in the coordinate system of the initial frame. Our focus is on improving reconstruction quality through data augmentation, without modifying the network architecture or training objectives, to isolate the effect of the augmentation itself. To this end, we propose two strategies for augmenting input video sequences: (i) *Image-to-Clips* (Section 3.1): synthesizes video clips with simulated geometry and camera motion from a single image. (ii) *Clips-to-Clips* (Section 3.2): extends augmentation to entire videos by generating diverse sub-clips that improve spatial coverage.

### 3.1 Image-to-Clips

A key property of video is the spatial overlap between consecutive frames, which induces multi-view consistency, a crucial element in 3D reconstruction. We use this principle to synthesize pseudo-video sequences from a single RGB-D image. Below, we detail the Image-to-Clips method (see Figure 2).

**Patch generation.** To simulate video-like continuity, we extract an ordered set of overlapping patches from a single RGB-D frame. As shown in Figure 2.B ("Generate patches"), each patch overlaps with at least one previous frame, mimicking temporal coherence. All generated patches share identical

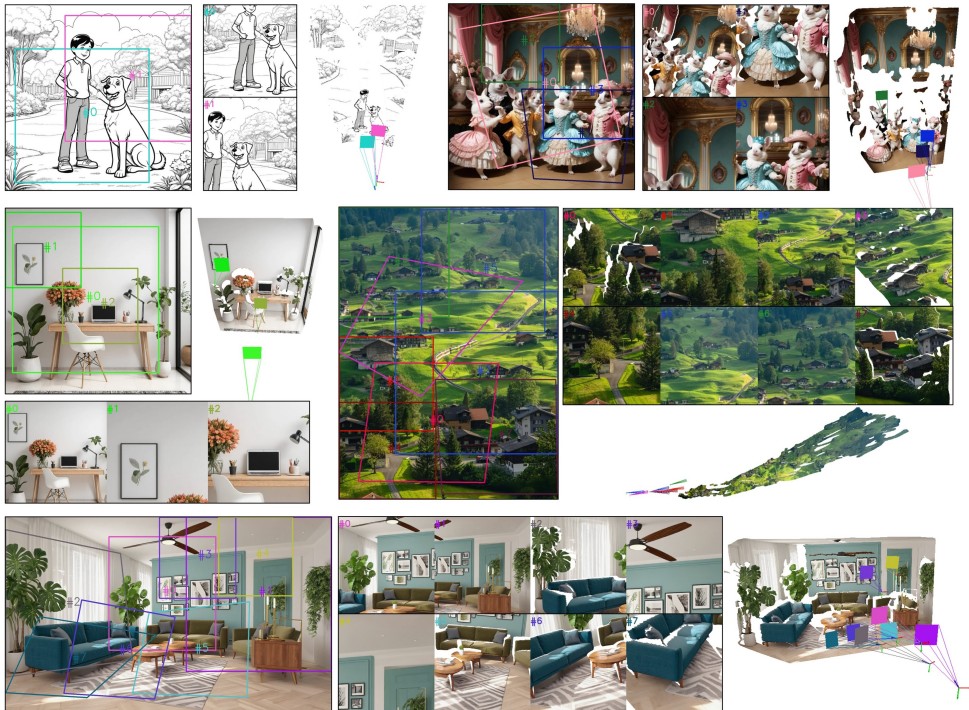

Figure 3: **Example of Image-to-Clips.** Given a single input image (left of each block), the proposed Image-to-Clips method samples ordered, overlapping patches (colored boxes) and assigns simulated 6-DoF camera trajectories (frustums) to generate view-consistent video clips. Examples across human-centric illustrations, indoor environments, and outdoor landscapes demonstrate the method's ability to synthesize diverse and realistic training sequences from a single frame.

resolution. Patch size determines the field of view: smaller patches mimic close-up views rich in local detail, while larger ones simulate wide-angle views, replicating zoom-in/out effects. We control overlap using bounding box IoU and dynamically vary the overlap ratio, starting high to promote consistency, then decreasing it to increase diversity and training difficulty. Whether ground-truth or predicted depth is used, we observe similar benefits (Figure 2.A uses predicted depth).

**Patch calibration.** We estimate the camera intrinsics and extrinsics for each patch, allowing for consistent geometry and downstream use. There are two calibration strategies. 1. *Varying intrinsics, fixed extrinsics*: modify the full-image intrinsics per patch for fast rendering, but cannot simulate camera motion. 2. *Fixed intrinsics, varying extrinsics*: use a constant intrinsics to emulate camera motion (Figure 3), although extrinsic estimation incurs errors. We employ a hybrid of the two approaches, with the second method being applied more frequently during training. Details follow.

*Varying intrinsics, fixed extrinsics.* In this case, $\mathbf{T}_{\text{w2c}}^m = \left[ I_3 \,|\, \mathbf{0} \right]$. Let the full-image intrinsics be $\mathbf{K}_{\text{full}} \in \mathbb{R}^{3\times3}$, and the depth map be $D \in \mathbb{R}^{H \times W}$. We define the corresponding 3D point map as:

$$X_{u,v} = \mathbf{K}_{\text{full}}^{-1} \left[ uD_{u,v}, vD_{u,v}, D_{u,v} \right]^\top, \tag{1}$$

where $(u, v) \in \{1, \ldots, W\} \times \{1, \ldots, H\}$ are pixel coordinates.

From each image, we extract $M \in \mathbb{Z}_{\geq 1}$ rectangular patches $\{B^m\}_{m=1}^M$, where $B^m = [u_1^m, v_1^m, u_2^m, v_2^m]^\top$ are the pixel coordinates of the top-left and bottom-right corners of the $m$-th patch. Each patch is resized to a fixed resolution $W' \times H'$. The intrinsics of each patch $\mathbf{K}_{\text{patch}}^m$ are derived from $\mathbf{K}_{\text{full}}$ via scaling:

$$\mathbf{K}_{\text{patch}}^m = \begin{pmatrix} f_x \, s_x^m & 0 & (c_x - u_1^m) \, s_x^m \\ 0 & f_y \, s_y^m & (c_y - v_1^m) \, s_y^m \\ 0 & 0 & 1 \end{pmatrix}, \quad s_x^m = \frac{W'}{u_2^m - u_1^m}, \quad s_y^m = \frac{H'}{v_2^m - v_1^m}. \tag{2}$$

*Fixed intrinsics, varying extrinsics.* Let consistent intrinsics per patch: $K_{\text{patch}}^m = \begin{pmatrix} f_x & 0 & W'/2 \\ 0 & f_y & H'/2 \end{pmatrix}$, to recover the extrinsic parameters $\mathbf{T}_{\text{w2c}}^m = [\mathbf{R}^m \mid \mathbf{t}^m]$ for each patch $m$, we first crop the dense point map to obtain a patch-specific 3D point set $X^m \in \mathbb{R}^{H' \times W' \times 3}$. We then estimate the camera pose by solving a Perspective-n-Point (PnP) problem using RANSAC [41, 42], which provides robustness to outliers. We define a set of 3D-2D correspondences $\{(X_i, x_i)\}_{i=1}^N$, where $X_i \in \mathbb{R}^3$ is a 3D point in world coordinates (from the depth map), and $x_i \in \mathbb{R}^2$ is its corresponding 2D location in the image patch. To estimate the rotation $\mathbf{R}^m \in SO(3)$ and translation $\mathbf{t}^m \in \mathbb{R}^3$, we minimize the total reprojection error over a set of inliers $\mathcal{I} \subseteq \{1, \ldots, N\}$.

This is formulated as:
$$\mathbf{R}^m, \mathbf{t}^m = \arg\min_{\mathbf{R}, \mathbf{t}} \sum_{i \in \mathcal{I}} \left\| \mathbf{x}_i - \pi \left( \mathbf{K}_{\text{patch}}^m \cdot (\mathbf{R} X_i + \mathbf{t}) \right) \right\|^2. \tag{3}$$

The projection function $\pi : \mathbb{R}^3 \to \mathbb{R}^2$ is defined as: $\pi([X, Y, Z]^\top) = \left( \frac{X}{Z}, \frac{Y}{Z} \right)^\top$. This process yields the optimal camera-to-world transformation for each patch, enabling accurate alignment between 2D image observations and 3D geometry.

**Camera motion augmentation.** After recovering geometric and camera parameters for each patch, we obtain all the necessary information to support standard 3D reconstruction and pose estimation tasks. However, this setup has a key limitation: simply cropping image patches mostly simulates translational motion, such as lateral or forward shifts, without introducing any realistic rotational dynamics (see Figure 2.C). Experimental results in supplementary materials show that combining both rotational and translational motion improves performance.

*Centroid-based camera rotation.* To simulate more realistic and diverse camera motion, we apply a controlled random rotation around the centroid of the 3D point cloud within each patch. Let the original camera-to-world pose be $\mathbf{T}_{\text{c2w}}^m = \begin{pmatrix} \mathbf{R}^{m\top} & -\mathbf{R}^{m\top}\mathbf{t}^m \\ \mathbf{0}^\top & 1 \end{pmatrix}$, and define the centroid of the 3D points as $\mathbf{c}_p = \frac{1}{N} \sum_{i=1}^N \mathbf{X}_i$. We sample a random rotation angle from a mixture of a uniform distribution Gaussian noise, $\theta \sim \mathcal{U}(\theta_{min}, \theta_{\max}) + \mathcal{N}(0, \sigma^2)$, and a random unit axis, $\mathbf{n} \sim \mathcal{N}(0, I)$, normalized such that $\|\mathbf{n}\| = 1$. Using Rodrigues' formula [43], we construct the corresponding rotation matrix $R_{\mathbf{n},\theta} \in SO(3)$ as:
$$\mathbf{R}_{\mathbf{n},\theta} = \cos\theta \, I + \sin\theta \, [\mathbf{n}]_\times + (1 - \cos\theta) \, \mathbf{n}\mathbf{n}^\top, \tag{4}$$

where $[\mathbf{n}]_\times$ is the skew-symmetric matrix of $\mathbf{n}$. The resulting 4×4 homogeneous transformation about the centroid $\mathbf{c}_p$ is:
$$\mathbf{T}_{\text{rot}} = \begin{pmatrix} \mathbf{R}_{\mathbf{n},\theta} & \mathbf{c}_p - \mathbf{R}_{\mathbf{n},\theta}\mathbf{c}_p \\ \mathbf{0}^\top & 1 \end{pmatrix}. \tag{5}$$

We update the camera pose by applying this rotation, and compute the new world-to-camera extrinsic matrix as:
$$\mathbf{T}_{\text{w2c}}'^m = (\mathbf{T}_{\text{rot}} \cdot \mathbf{T}_{\text{c2w}}^m)^{-1}. \tag{6}$$
This augmentation introduces realistic viewpoint variation while maintaining the scene focus, enabling free-viewpoint rotation beyond what patch cropping can achieve, as shown in Figure 2.

*Valid view checks.* Random rotation can result in invalid camera views that negatively impact training. We therefore apply two geometric validation tests to ensure the quality of the rotated view: *front-surface coverage test* and *view-frustum inclusion test*.

   i. *Front-surface coverage test.* This test evaluates whether the camera observes a sufficient amount of front-facing geometry. For each 3D point with position $p_i$ and outward-facing unit normal $n_i$, we compute the dot product between the normal and the viewing direction from the camera center, given by $\mathbf{c}_c = -\mathbf{R}'^{m\top}\mathbf{t}'^m$. The front-surface coverage score is defined as:
$$\text{FrontCov} = \frac{1}{N} \sum_{i=1}^N \mathbf{1}\left( \mathbf{n}_i^\top \frac{(\mathbf{c}_c - \mathbf{p}_i)}{\|\mathbf{c}_c - \mathbf{p}_i\|_2} > \cos\theta_{\text{valid}} \right), \tag{7}$$

where $\theta_{\text{valid}}$ is a predefined angular threshold, $\mathbf{1}(\cdot)$ is the indicator function, $N$ is the number of 3D points, $\mathbf{p}_i \in \mathbb{R}^3$ is the 3D position of point $i$, and $\mathbf{n}_i \in \mathbb{R}^3$ is the corresponding unit normal

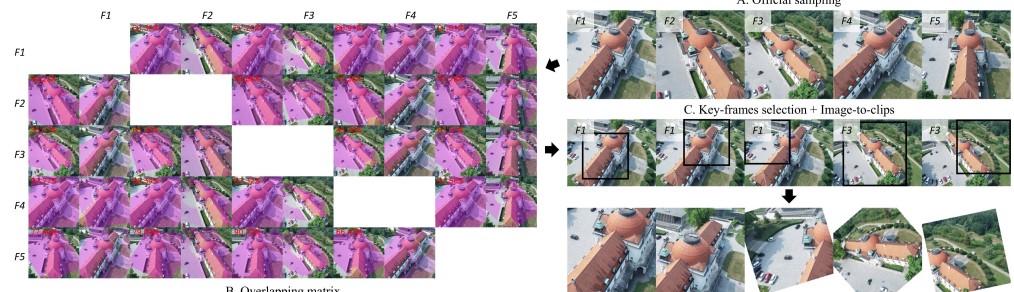

Figure 4: **Puzzles: Clips-to-Clips.** (**A**) We begin by uniformly sampling frames from a video. (**B**) A pair-wise overlap matrix is computed to measure frame redundancy, with overlap visualized in purple and overlap ratios annotated in red. (**C**) Low-redundancy keyframes are then selected, and diverse sub-clips are synthesized from them using the Image-to-Clips method.

vector. The metric $\mathrm{FrontCov} \in [0, 1]$ quantifies the fraction of points that are front-facing with respect to the camera. A score of $\mathrm{FrontCov} = 1$ indicates all points are front-facing, while $\mathrm{FrontCov} = 0$ implies that none of the points face the camera.

ii. *View-frustum coverage test.* This test ensures that the projected 3D points lie within the visible image frame. Each 3D point $\mathbf{p}_i$ is projected into the image plane using the updated camera extrinsics and the patch-specific intrinsic matrix $\mathbf{K}_{\mathrm{patch}}^m$:

$$\begin{pmatrix} u_i \\ v_i \\ 1 \end{pmatrix} \sim \mathbf{K}_{\mathrm{patch}}^m (\mathbf{R}'^m \mathbf{p}_i + \mathbf{t}'^m), \ g_i = \mathbf{1}\big(0 \le u_i < W', \ 0 \le v_i < H', \ D_{u_i, v_i} > 0\big), \quad (8)$$

where $g_i$ is a binary indicator that equals 1 if the projected point falls within the image bounds and has a positive depth value. We define the image coverage score as: $\mathrm{ImgCov} = \frac{1}{H'W'} \sum_i g_i$, which measures the fraction of the projected points that are valid and visible in the image.

During augmentation, we generate a finite set of candidate camera poses and select the one that maximizes both $\mathrm{FrontCov}$ and $\mathrm{ImgCov}$. Finally, we render the point clouds using Open3D [44] with the augmented camera parameters to generate high-quality, posed video-depth sequences (see bottom row of Figure 2). If camera rotation reveals occluded areas, we fill them with white; if too many occlusions occur, we discard that patch's rotation. The main text focuses on camera motion augmentation, while additional 2D augmentations are detailed in the supplementary material.

### 3.2 Clips-to-Clips

To extend coverage beyond individual images, we generalize our augmentation strategy from Image-to-Clips to Clips-to-Clips. Video clips often exhibit significant redundancy (see Figure 1.A), and prior methods [9, 10] typically sample frames at fixed or random intervals, overlooking the spatial relationships between them (details in supplementary material). This can result in pointmaps with little or no overlap between views, leading to geometrically inconsistent clips that hinder training and reduce accuracy. Our goal is to select key frames that both preserve scene continuity and provide sufficient visual coverage. We then apply the Image-to-Clips augmentation on these selected frames.

**Overlapping matrix.** Given the camera parameters and depth maps, we compute geometric overlap between video frames using an overlapping matrix $O \in \mathbb{R}^{N \times N}$, where $N$ is the total number of frames. Each element $O_{ij} \in [0, 1]$ quantifies the fraction of overlapping geometry between frames $i$ and $j$. A value of $O_{ij} = 0$ means no overlap, while $O_{ij} = 1$ indicates full overlap. As shown in Figure 4.B, this matrix provides a principled basis for selecting a minimal yet representative set of key frames that maximize coverage while minimizing redundancy. To compute $O_{ij}$, we treat each frame $v^s$ as a source view and project its pixels into a reference view $v^r$. Let $\mathbf{x}^s$ be the homogeneous coordinates of a pixel in the source frame, and $D^s(\mathbf{x})$ its depth. We compute the reprojected coordinates $\mathbf{x}^{s \rightarrow r}$ and the corresponding depth $\mathbf{x}^s$ in the reference frame. We then compute the reprojected depth with the reference depth map $D^{s \rightarrow r}$ at the same pixel location. The

overlap score is defined as:

$$O_{ij} \;=\; \frac{1}{|\mathcal{V}_x|} \sum_{x \in \mathcal{V}_x} \mathbf{1}\Big( \big| D^{s \to r}(\mathbf{x}^{s \to r}) - D^r(\mathbf{x}^{s \to r}) \big| < \tau \Big), \qquad (9)$$

where $\mathcal{V}_x$ is the set of valid source pixels ($D_x > 0$) and $\tau$ is a threshold that accounts for scene-dependent depth variance.

> **Key-frame selection.** To ensure comprehensive scene coverage with minimal redundancy, we extract representative frames from the overlap matrix $O$ using a three-step procedure (see Figure 5):
>
> i. **Validity filtering**: Retain only frames whose maximum overlap with any other frame exceeds a threshold $\eta$: $\mathcal{V}_f = \{\, i \mid \max_{j \neq i} O_{ij} > \eta \,\}$. For example, the gray frame in Figure 5(A), with an overlap 0.1, is discarded.
>
> ii. **Longest cover set**: Among valid frames, select a seed frame $S$ that has the largest number of neighbors with sufficient overlap: $S = \arg\max_{i \in \mathcal{V}_f} \big| \{\, j \in \mathcal{V}_f \mid O_{ij} \geq \tau \,\} \big|$. In Figure 5(B), the purple frame is selected over the pink on as it covers three neighbors instead of two.
>
> iii. **Redundancy pruning**: From this set, keep frame $j$ only if it is not highly redundant with any earlier selected frame: $P = \{\, j \in S \mid \forall k < j,\ \max(O_{jk}, O_{kj}) < \rho \,\}$. For instance, the pink frame is excluded in Figure 5(C) due to strong bidirectional overlap with an earlier frame.

The final keyframe set $P$ removes isolated frames, identifies the largest group of mutually overlapping views, and eliminates redundant selections As shown in Figure 4, the original clip contains repeated views of both the front and back of the building. Our method selects two representative frames: front ($F_3$) and back ($F_1$), as keyframes, and applies Image-to-Clips augmentation to each, resulting in more diverse and informative training clips.

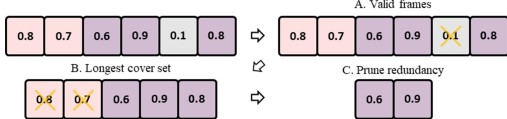

Figure 5: **Keyframe selection.** Colors represent overlapping groups; numbers indicate pairwise overlap scores $O_{ij}$. We discard low-overlap frames, keep the highest-coverage group, and remove redundant ones to form the final keyframes.

**Plug-and-play augmentation.** Our Clips-to-Clips is modular and easily integrates into most modern 3D reconstruction frameworks. It accepts standard video clips and outputs augmented, geometry-aware training samples. The key-frame selection process is internal and does not require altering the original data loading or sampling logic.

## 4 Experiments

### 4.1 Setup

**Datasets and metrics.** We train on a blended corpus comprising ScanNet-v2 [12], ARKitScenes [45], a selected Habitat subset [46], and the in-the-wild/object dataset BlendedMVS [47], totaling approximately **14 million** images. For evaluations in Sections 4.2 and 4.3, we draw uniform subsamples of varying size to study the scaling behavior of Puzzles. To assess the impact of our augmentation, we evaluate on three unseen datasets: 7Scenes [48], NRGBD [49] and DTU [50] using Accuracy (Acc) and Completion (Comp) metrics from [9]. Predicted dense point maps are compared to back-projected ground-truth depth , excluding invalid and background points.

**Baselines.** We adopt three representative video-based *3R*-series methods, Spann3R [9], SLAM3R [10], and Fast3R [11], as baselines. Since each was originally trained on a distinct dataset, we preserve their published training protocols and architectures, and retrain them on a unified dataset both *with* and *without* our Puzzles data augmentation. As a result, the reported metrics may deviate from those reported in the original papers; our objective is not to replicate prior results, but to investigate how each model responds to changes in data distribution. Using Spann3R as an example, we follow the official training setup and train for 120 epochs on the same dataset with the same 14M training samples. The only change is the application of Puzzles augmentation. Puzzles is a plug-and-play module applied on-the-fly through the dataloader, without requiring any pre-processing or modification of the training pipeline. Training was conducted on eight H100 GPUs for 120 epochs. The total training time was approximately 40 hours with Puzzles, compared to 36 hours without. We

| Method | w/ Puzzles | Data | 7Scenes | | | | NRGBD | | | | DTU | | | |
|---|---|---|---|---|---|---|---|---|---|---|---|---|---|---|
| | | | Acc↓ | | Comp↓ | | Acc↓ | | Comp↓ | | Acc↓ | | Comp↓ | |
| | | | Value | Δ(%) | Value | Δ(%) | Value | Δ(%) | Value | Δ(%) | Value | Δ(%) | Value | Δ(%) |
| **Spann3R** [9] | | full | 0.0388 | | 0.0253 | | 0.0686 | | 0.0315 | | 6.2432 | | 3.1259 | |
| | ✓ | 1/10 | 0.0389 | −0.26 | 0.0248 | +1.98 | 0.0753 | −9.79 | 0.0341 | −8.50 | 4.9832 | +20.18 | 2.5172 | +19.47 |
| | ✓ | full | 0.0330 | +14.94 | 0.0224 | +11.46 | 0.0644 | +6.00 | 0.0291 | +7.51 | 5.0004 | +19.90 | 2.5113 | +19.66 |
| **Fast3R** [11] | | full | 0.0412 | | 0.0275 | | 0.0735 | | 0.0287 | | 4.2961 | | 2.0681 | |
| | ✓ | 1/10 | 0.0402 | +2.30 | 0.0272 | +1.09 | 0.0772 | −5.11 | 0.0295 | −2.78 | 3.7174 | +13.47 | 1.8941 | +8.41 |
| | ✓ | full | 0.0342 | +16.99 | 0.0239 | +13.09 | 0.0684 | +6.94 | 0.0259 | +9.75 | 3.5912 | +16.41 | 1.7379 | +15.96 |
| **SLAM3R** [10] | | full | 0.0291 | | 0.0245 | | 0.0481 | | 0.0292 | | 4.3820 | | 2.4754 | |
| | ✓ | 1/10 | 0.0289 | +0.68 | 0.0237 | +3.26 | 0.0493 | −2.49 | 0.0313 | −7.19 | 3.5980 | +17.89 | 2.0891 | +15.60 |
| | ✓ | full | 0.0264 | +9.27 | 0.0218 | +11.02 | 0.0439 | +8.73 | 0.0263 | +9.93 | 3.6497 | +16.71 | 2.0762 | +16.12 |

Table 1: Quantitative comparison on 7Scenes [48], NRGBD [49] and DTU [50]. For each metric we report both the raw value and the relative improvement (△) achieved by incorporating Puzzles. All methods are evaluated on long video sequences without bundle adjustment refinement; note that Spann3R operates in online mode.

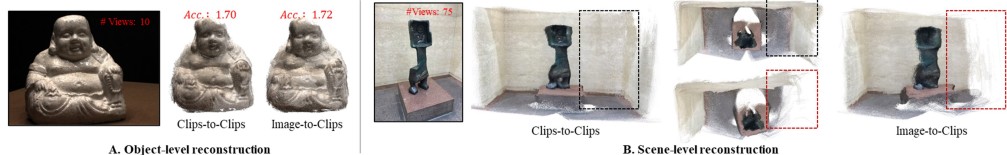

Figure 6: **Comparison of Clips-to-Clips and Image-to-Clips. A**. Object-level reconstruction using 10 input views ($\#Views = 10$), where both methods perform comparably. **B**. Scene-level reconstruction with 75 input views ($\#Views = 75$), where Image-to-Clips exhibits drift and unstable predictions, resulting in the discard of low-confidence outputs.

observed slightly slower convergence when using Puzzles, which is expected due to the increased diversity and complexity of the training data. While longer training could yield further improvements, we use the same configuration across experiments to ensure fair comparison.

**Implementation details.** To generate $N$ patches from a single image, the first patch is generated randomly. Each subsequent patch is then created to ensure it overlaps with at least one of the previously generated patches, based on IoU calculations. This overlap constraint helps maintain spatial continuity and supports consistency across views. The patch size is determined dynamically, based on the image resolution and the desired number of patches per image. In addition, each newly generated patch is required to overlap with at least one of the previously generated patches, but not necessarily with consecutive frames. This design choice accommodates scenarios with rapid camera motion, where adjacent frames may have little or not overlap.

For Clips-to-Clips, we employ a cut-off threshold $\eta = 0.1$, a minimum visibility threshold $\tau = 0.2$, and a redundancy threshold $\rho = 0.7$. In Image-to-Clips, we constrain the maximum angle of incidence to $\theta_{\text{valid}} = 100°$, and sample rotation angles $\theta$ uniformly from $[30°, 90°]$. Additionally, we apply a random affine perturbation to the pointmaps: rotations up to $\pm 45°$, translations up to 20% of the bounding box size, and scaling factors uniformly sampled from $[0.8, 1.0]$. The official source code for our method is available on GitHub: https://github.com/Jiahao-Ma/puzzles-code.

## 4.2 Evaluation

**Scene-level reconstruction.** On scene-level 7Scenes and NRGBD benchmarks, integrating Puzzles consistently improves accuracy across all *3R*-series methods. Fast3R shows the largest gains on 7Scenes with 17.0% reduction in Acc and 13.1% in Comp, while Spann3R and SLAM3R also see notable improvements. On NRGBD, gains are smaller but stable ($6 \sim 10\%$). Notably, all models trained with just $1/10$ of the training data match or surpass their full-data baselines on 7Scenes, though not on NRGBD. This discrepancy likely stems from NRGBD's larger and more complex scenes, which are more prone to drift and demand greater data diversity. These results demonstrate that Puzzles enhances generalization and robustness, especially in large-scale reconstruction, without requiring architecture modifications.

**Object-level reconstruction.** On the object-level DTU benchmark, Puzzles yields consistently large improvements across all methods. SLAM3R benefits the most, with a 16.71% reduction in accuracy

error and a 16.12% reduction in completion error. Spann3R and Fast3R also improve substantially, each showing gains of $16 \sim 20\%$ in accuracy. Notably, our training data differs from that used in [9–11], as we do not incorporate a dedicated object-level dataset [14]–only a small portion of BlendedMVS is included. Despite this, Puzzles effectively generates diverse and detailed local data, resulting in strong performance gains on DTU.

## 4.3 Analysis

**Camera pose distribution after puzzles.** We analyze the difference in camera pose distribution across the dataset used in the main paper.

As shown in Table 2, Puzzles follow symmetric zero-mean augmentation that leads to no obvious changes in mean. The Puzzles augmentation introduces pronounced variance increases across all pose axes. In the original dataset, broader axes ($yaw$, $t_x$, $t_y$) exhibit only modest variance gains, whereas previously narrow axes ($pitch$, $roll$, $t_z$) expand substantially.

| Statistic | roll (°) | pitch (°) | yaw (°) |
|---|---|---|---|
| Rotation $\mu$ | $-89.6$ | $0.04$ | $-4.7$ |
| **Statistic** | $t_x$ (m) | $t_y$ (m) | $t_z$ (m) |
| Translation $\mu$ | $-1.58$ | $-1.39$ | $2.45$ |

Table 2: Mean camera poses ($\mu$). Zero-mean augmentations in Puzzles preserve pose symmetry.

After augmentation, as shown in Table 3, most rotation axes fall within the $50° \sim 110°$ range, and translations cluster around $20 \sim 35$ m. This yields a more balanced and comprehensive pose distribution, exposing the model to a wider range of viewpoints. The enlarged standard deviations ($\sigma$) quantitatively capture the degree to which Puzzles expands the pose space, thereby enhancing generalization to rare viewpoints, improving robustness to atypical camera motions, and ultimately boosting model performance.

| Axis | $\sigma_{\text{orig}}$ | $\sigma_{\text{new}}$ | **Increase** |
|---|---|---|---|
| Rotation X (roll) | 15.9° | 54.3° | +242% |
| Rotation Y (pitch) | 5.5° | 52.2° | +853% |
| Rotation Z (yaw) | 93.7° | 107.2° | +14% |
| Translation $t_x$ | 31.1 m | 35.6 m | +14.4% |
| Translation $t_y$ | 30.9 m | 35.4 m | +14.6% |
| Translation $t_z$ | 10.2 m | 20.1 m | +96.5% |

Table 3: Camera-pose standard deviation ($\sigma$) before and after augmentation.

**Impact of white filling.** The Image-to-Clips process inevitably introduces occluded regions due to camera motion. These regions are filled with white color and excluded from supervision during optimization. We explore the impact of the occluded regions. Specifically, we experimented with the inpainting method LAMA [51] to fill the occluded regions, while

| Setting | Acc↓ | Comp↓ | **Time ↓ [h]** |
|---|---|---|---|
| Spann3R (w/o. Inpainting) | 0.0330 | 0.0224 | 40 |
| Spann3R (w/. Inpainting) | 0.0332 | 0.0221 | 53 |
| $\triangle$(%) | −0.6 | +1.3 | +32.5(+13h) |

Table 4: Effect of inpainting: increased training time with negligible accuracy change.

excluding the inpainted areas from the loss computation. However, as shown in Table 4, our experiments conducted on eight NVIDIA H100 GPUs indicate that inpainting offers no significant performance improvement, while increasing the training time by approximately 32.5%. Moreover, existing inpainting approaches often introduce noticeable artifacts when filling large missing areas, which can contaminate valid image regions. Therefore, we chose to discard this strategy in our final pipeline.

**Component effect analysis.** As shown in Table 5, We further explore the ablation study to illustrate the impact of each component. The experiments show that Clips-to-Clips (C2C) bring the most significant performance gains, while Image-to-Clips (I2C) with augmentation (w/. Aug.) provides a smaller but still positive improvement. In scene-level reconstruction, I2C with augmentation helps simulate camera motion, allowing the network to capture local de-

| Config. | Components | | | 7Scenes | |
|---|---|---|---|---|---|
| | I2C w/o. Aug. | I2C w/. Aug. | C2C | Acc↓ | $\triangle$(%) |
| Baseline | ✓ | | | 0.1278 | — |
| + Aug. | ✓ | ✓ | | 0.0916 | +28.3 |
| + C2C | ✓ | ✓ | ✓ | **0.0330** | +64.0 |

Table 5: Impact of Image-to-Clips (I2C), Clips-to-Clips (C2C), and augmentation (Aug.) components on 7Scenes.

tails. In contrast, C2C improves robustness and supports large-scale reconstruction by capturing global context.

**Image-to-Clips *vs.* Clips-to-Clips.** Image-to-Clips generates dense, posed video-depth sequences but inherently offers only local spatial coverage. Clips-to-Clips generalizes this idea by supplying significantly broader scene-level coverage. As illustrated in Figure 6.A, and the DTU results in

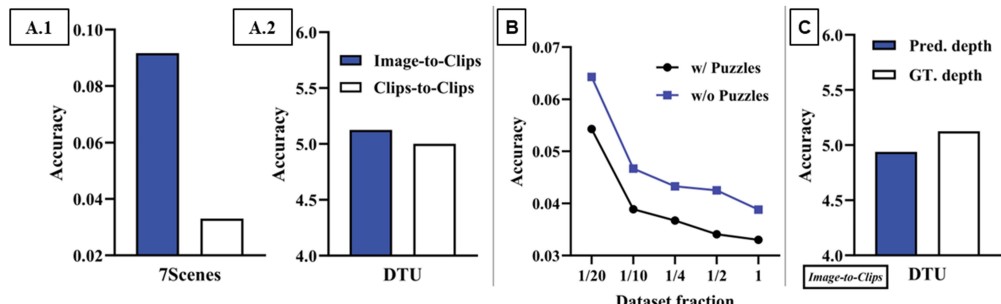

Figure 7: **Ablation study. A**. Comparison on 7Scenes [48] (scene-level) and DTU [50] (object-level). **B**. Model accuracy with Puzzles at varying data fractions. **C**. Image-to-Clips single-image augmentation: model accuracy comparing predicted vs. ground-truth depth. All experiments use Spann3R as the baseline, and accuracy is reported using the same metric throughout (lower is better). The results in sub-figure **B**. are based on Clips-to-Clips while **C**. corresponds to Image-to-Clips.

Figure 7.A.2, both methods attain comparable accuracy for object-level reconstruction. However, in full scene reconstruction, as shown in Figure 6.B and 7Scenes results in Figure 7.A.1, Clips-to-Clips produce better and more stable results than Image-to-Clips. This is because Clips-to-Clips supplies the network with additional scene-level supervision that Image-to-Clips lacks. This broader coverage mitigates drift and stabilizes the inference, especially when test sequences contain more frames than seen during training.

**Model accuracy across increasing dataset fractions.** The experiment illustrated in the middle of Figure 7.B, compares Puzzles augmentation (black) to the baseline without augmentation (blue), over data fractions ranging from $1/20$ to full. Lower values indicate better performance. Puzzles consistently yields gains at all fractions, with the largest relative gain at $1/20$ (over **15**% improvement). Even with full data, Puzzles reduces error from 0.0388 to 0.0330 (see Table 1), demonstrating improved generalization. Notably, models trained with Puzzles on just $1/10$ of the data match or exceed the performance of full-data baselines. Beyond $1/10$, Puzzles consistently outperforms training on the entire dataset without augmentation.

**Robustness of Image-to-Clips to predicted *vs.* ground-truth depth.** We evaluate the sensitivity of Image-to-Clips augmentation to the quality of depth input using the DTU benchmark, as shown in Figure 7.C. Two training settings are contrasted: one where augmented clips are paired with predicted depth from an off-the-shelf monocular estimator [15] (solid blue bar), and another using the dataset's ground-truth depth (white bar). The model trained with predicted depth achieves an accuracy of roughly 4.9, marginally outperforming the 5.1 result with ground-truth depth. This small difference, well within experimental variance, demonstrates that Puzzles is robust to depth quality and remains effective even without precise supervision, enabling the method to scale to large, unlabeled image collections without sacrificing reconstruction quality, reinforcing its "unbounded" scalability.

**Trends and challenges.** Video-based *3R*-series methods show strong potential for dense 3D reconstruction, but they tend to suffer from drift when scaling to large scenes with more views than seen during training. One mitigation strategy [52] combines learned front-end pointmap prediction with a global optimization back-end. A more radical alternative [9, 11, 10] adopts a fully data-driven approach, requiring a scalable architecture and vast scene-level datasets. In contrast, we pursue a data-centric solution: our current augmentation is scene-specific, but future work could explore stitching clips across scenes to synthesize unbounded, large-scale video-depth data.

## 5 Conclusion

We presented Puzzles, a plug-and-play data-augmentation framework that transforms single images or short clips into high-quality, unbounded video-depth sequences through ordered, overlapping patch synthesis. When integrated into existing *3R*-series pipelines, Puzzles consistently boosts reconstruction accuracy Remarkably, models trained with just **10**% of the original data, augmented with Puzzles, can match or exceed the performance of full-data baselines, demonstrating the power of a data-centric solution to scalable 3D reconstruction.

## Acknowledgments and Disclosure of Funding

We sincerely thank the anonymous reviewers for their invaluable insights and constructive feedback, which have greatly contributed to improving our work.

This work is supported by the CSIRO Data61 PhD Scholarship.

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

# A  Additional Examples of Clips-to-Clips

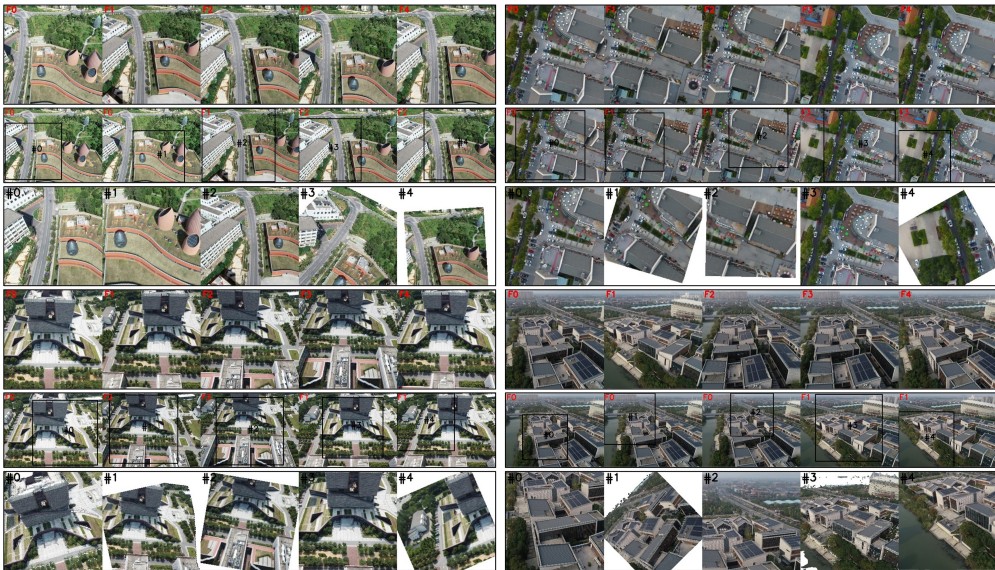

Figure 8: **Example of Clips-to-Clips.** *Top*: Consecutive frames from the original training clip. *Middle*: Selected Keyframes with corresponding patch selections. *Bottom*: Synthetic video clips generated from keyframes using the Image-to-Clips process.

We present additional examples of the Clips-to-Clips process in Figure 8, as a supplement to Figure 4 in the main text. *Top row*: These are consecutive frames extracted from a raw training video. The sequence exhibits high temporal redundancy, neighboring frames share nearly identical viewpoints and contribute minimal new geometric information. *Middle row*: Keyframe selection. We highlight the patches that maximize scene coverage using black bounding boxes and indices. These selected keyframes ensure continuous, dual-sided visibility of the target structure, capturing the most informative perspectives within the clip. *Bottom row*: Augmented clip via Image-to-Clips. Starting from the selected keyframes, the Image-to-Clips augmentation synthesizes new video sequences with greater baseline shifts, camera rotations, and scale variations. Despite these transformations, inter-frame overlap is preserved, ensuring geometric consistency. The resulting clips are significantly richer in structure and offer more diverse visual cues compared to the original sequence.

# B  Additional Experiments and Evaluations

## B.1  Joint 2D-3D Data Augmentation

In addition to simulating camera motion, we extend the standard PyTorch data augmentation pipeline to operate jointly on images, point clouds, and overlap masks, incorporating both affine and perspective transformations. Specifically, we apply a `RandomAffine` augmentation with rotation angles sampled from the range $[-45°, 45°]$, translations up to $20\%$ of the image dimensions, and scaling factors between $0.8$ and $1.0$. We also adopt `RandomPerspective` transformation with a distortion scale of $0.1$.

To maintain geometric plausibility and prevent excessive warping that could degrade 3D reconstruction quality, the `RandomPerspective` transformation is applied with a low probability. The visual effects of these augmentations are shown in the bottom rows of Figure 8.

Importantly, these 2D augmentations are applied only to paired images and their corresponding point clouds. We do not currently update the associated camera parameters to reflect the applied transformations, an omission that represents a potential direction for future work, particularly for enhancing geometric consistency in training.

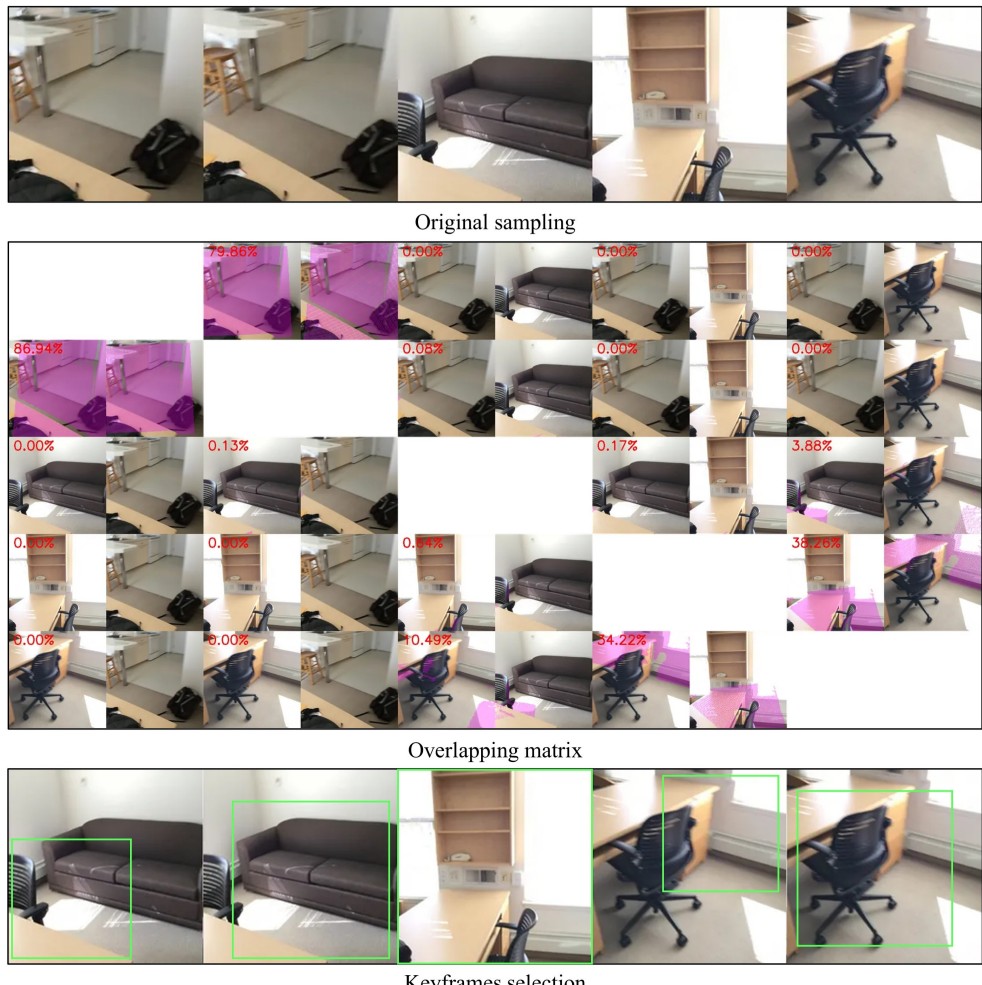

Figure 9: From random sampling to coverage-aware key-frame selection. **Top:** arbitrary sampling yields spatial discontinuities. **Middle:** overlap matrix shows pair-wise coverage percentages. **Bottom:** key-frame selection (green boxes) picks frames that maximise coverage and continuity.

| Method | w/ Puzzles | Data | Acc↓ | | Comp↓ | |
|--------|:---:|------|-------|------|-------|------|
| | | | Value | Δ(%) | Value | Δ(%) |
| **Spann3R** | | 1/10 | 0.0542 | — | 0.0325 | — |
| | ✓ | 1/10 | 0.0389 | +28.2 | 0.0248 | +23.7 |
| | | full | 0.0388 | — | 0.0253 | — |
| | ✓ | full | **0.0330** | +14.9 | **0.0224** | +11.4 |

Table 6: Quantitative comparison on 7Scenes. For each metric, both the raw value and the relative improvement (△) achieved by incorporating Puzzles are reported. The checkmark indicates configurations trained with Puzzles.

## B.2 Impact of Puzzles under Limited Data

We have supplemented the evaluation with an experiment where only 10% of the training data is used without Puzzles augmentation. The improvement shown in the Table 6 are based on the comparison under the same data size. Compared to the baseline using 10% of the data, applying Puzzles augmentation significantly improves reconstruction quality. Notably, under this limited data regime, the ACC and Comp errors are reduced by 28.2% and 23.7%, respectively. This demonstrates that Puzzles greatly improves data efficiency. While the original setting relies on large-scale data to achieve diversity, Puzzles maximizes the utility of available data by generating more informative and varied training samples.

## B.3 Keyframe Selection for Spatial Continuity

Existing video-based *3R*-series methods often sample clips by selecting frames at arbitrary intervals. This random sampling strategy can result in spatially discontinuous sequences, where frames lack sufficient overlap for effective 3D reconstruction.

Figure 9 illustrates how our keyframe selection strategy addresses this issue. In the top row ("original sampling"), five frames are randomly selected from a training clip. However, the first two frames share almost no common field of view with the remaining three, leading to poor geometric continuity. The middle grid shows the pairwise geometric overlap between these frames: red percentages and purple marks indicate high overlap between the first two frames (approximately 80%), near-zero overlap between those and the final three, and moderate overlap among the last three (up to $\sim 38\%$).

Using the overlapping matrix, our method identifies and discards the spatially disconnected first two frames, retaining only the final three as representative keyframes. These are shown in the botton row. The green rectangles indicate the image regions that will be cropped and warped during the subsequent Image-to-Clips augmentation. This ensures that the generated clip maintains sufficient scene coverage and smooth inter-frame continuity, both critical for learning effective 3D representations.

## B.4 Translation and Rotation Augmentation

As shown in Figure 10, camera translation-only augmentation (*TransAug*) involves simple cropping-based patch generation (Figure 2.B, "generate patches"), while *Trans+RotAug* extends this by applying centroid-based camera rotation around each generated patch (the full "Augmentation" pipeline in Figure 2.D). As shown in Figure 10, translation-only augmentation (*TransAug*) involves generating image patches through simple cropping (see main text Figure 2.B, "generate patches"). In contrast, translation-plus-rotation augmentation (*Trans+RotAug*) extends this approach by applying centroid-based camera rotations around each patch, corresponding to the full "Augmentation" pipeline shown in Figure 2.D of main text.

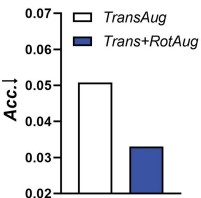

Figure 10: Reconstruction accuracy (lower Acc. is better) for translation-only augmentation (*TransAug*) versus combined translation and rotation augmentation (*Trans+RotAug*).

Quantitatively, combining translation and rotation yields the lowest mean reconstruction error of 0.0330, representing an improvement of approximately 35% over translation-only augmentation (TransAug, 0.0508). This substantial gain highlights the benefit of jointly perturbing both camera position and orientation. Not only does this increase the diversity of geometric viewpoints in training data, but it also leads to more robust and generalizable 3D reconstruction performance.

## B.5 Impact of Puzzles on 3D Reconstruction

Figure 11 presents a comparison of reconstruction results from three methods, SLAM3R, Fast3R, and Spann3R, both before (middle) and after (right) applying Puzzles. In all cases, applying Puzzles leads to a notable improvement in reconstruction quality by enhancing completeness and reducing noise and holes. Specifically, SLAM3R benefits from the filling of boundary gaps, Fast3R recovers

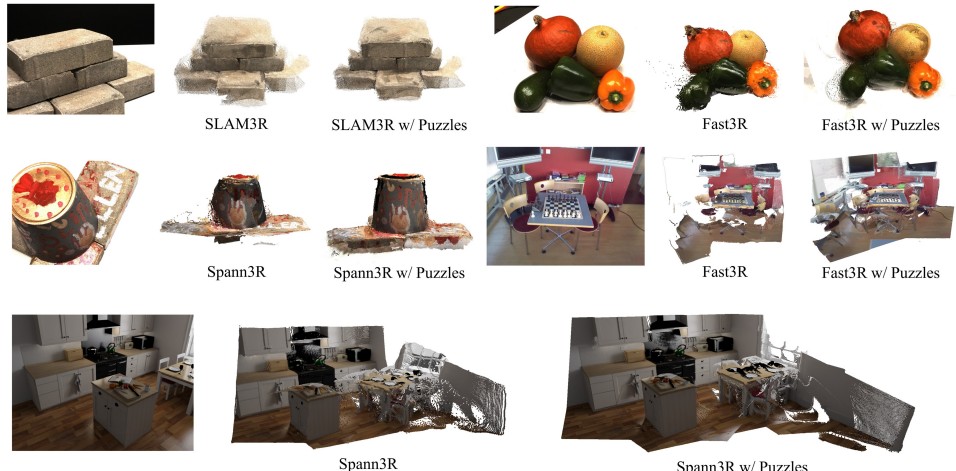

Figure 11: Reconstruction results without (left) and with (right) the proposed Puzzles augmentation.

| Setting | Acc↓ | Comp↓ |
|---|---|---|
| w/o. Aug. | 0.0395 | 0.0261 |
| Cropping + Jittering + Flipping | 0.0388 | 0.0253 |
| **Puzzles** | **0.0330** | **0.0224** |

Table 7: Comparison of different augmentation strategies.

finer details with fewer artifacts, and Spann3R more accurately reconstructs missing structures while producing smoother surfaces. Please refer to the project website for a detailed visual comparison.

### B.6 Comparing Puzzles with Traditional Augmentation

As show in Table 7, we compared the performance of Spann3R on the 7Scenes dataset under three settings: (1) no augmentation (w/o. Aug.), (2) traditional augmentation (cropping + jittering + flipping), and (3) our proposed Puzzles. In the official implementation, traditional augmentation includes cropping, jittering, and flipping, which enhance appearance diversity at the image level. In contrast, Puzzles augments data from a spatial perspective by simulating camera motion and introducing diverse viewpoints. Our results show that Puzzles leads to more effective generalization, outperforming both the no-augmentation and traditional augmentation settings.

## C Limitations and Potential Future Work

While Puzzles demonstrates strong performance gains in static scene reconstruction under the standard pinhole camera model, several limitations remain that present exciting avenues for future exploration:

i. **Support for diverse camera models.** Currently, Puzzles operates under the assumption of a pinhole camera model. Extending support to alternative camera types, such as fisheye, panoramic, or catadioptric lenses, would enable augmentation for a broader range of real-world capture devices and use cases, especially in robotics and immersive applications.

ii. **Dynamic scene augmentation.** The current framework assumes scene rigidity and does not account for object or human motion within a frame. Future work could explore strategies for augmenting dynamic scenes, for example, by learning to synthesize plausible motion trajectories or segmenting and animating scene components independently.

iii. **Cross-scene composition and stitching.** Puzzles presently operates on single scenes or short clips. An extension to stitch patches or sub-clips across multiple distinct scenes could simulate transitions between rooms or outdoor environments, enriching training diversity for large-scale 3D reconstruction tasks. This could also help bridge domain gaps between datasets.

iv. **Learning-guided augmentation.** Currently, Puzzles applies geometric transformations via hand-designed procedures. Incorporating learning-based augmentation policies, where the system automatically optimizes transformations to maximize reconstruction utility, could further improve performance and generalization.

v. **Physics-aware realism.** Enhancing the photometric and geometric fidelity of augmentations (*e.g.*, simulating lighting changes, motion blur, or depth-dependent occlusions) could bring synthetic data even closer to real-world conditions, leading to more robust models in challenging environments.

These directions suggest that Puzzles can evolve beyond a data augmentation strategy into a more general framework for synthetic 3D scene generation and learning. We view this work as a foundation toward that broader goal.

## D   Broader Impacts

The development of Puzzles introduces both promising opportunities and important considerations for the broader computer vision and 3D reconstruction communities.

**Positive impacts.** By significantly reducing the dependency on large-scale posed video-depth datasets, Puzzles democratizes access to high-quality 3D reconstruction tools. Researchers and practitioners in resource-constrained settings, such as those lacking access to expensive sensors or extensive capture setups, can now generate effective training data from minimal inputs. This may accelerate progress in areas like robotics, AR/VR, digital heritage preservation, and autonomous navigation, particularly in domains where data collection is difficult or expensive (*e.g.*, remote environments, cultural landmarks, or historical archives). The plug-and-play nature of Puzzles also encourages wider adoption across various architectures, facilitating rapid experimentation and innovation in data-efficient 3D learning.

**Risks and ethical considerations.** As with many advancements in generative data augmentation, the increased realism and diversity of synthetic training samples also raise potential concerns. For example, improved 3D reconstruction capabilities could be misused for surveillance, privacy-invasive applications, or the creation of deepfakes. While Puzzles itself does not introduce adversarial manipulation, it lowers the barrier to creating convincing 3D models from limited imagery, which could be repurposed in harmful ways.

To mitigate these risks, we encourage responsible deployment, including transparency around the use of synthetic data, and advocate for integration with ethical guidelines when applying this method in sensitive domains. We also recommend further exploration of watermarking or provenance tracking techniques for synthetic data generated via Puzzles.

**Environmental considerations.** By enabling more data-efficient training pipelines, Puzzles may reduce the computational cost and environmental footprint associated with large-scale 3D model training. Training models on only 10% of the original dataset, while achieving similar performance, can substantially cut energy usage, aligning with broader goals of sustainable AI research.

