# OpenReview forum: "Puzzles: Unbounded Video-Depth Augmentation for Scalable End-to-End 3D Reconstruction"
_NeurIPS.cc/2025/Conference — NeurIPS 2025 poster_

### Official Review · Reviewer_SXr5 · 2025-07-01

**Clarity:** 2
**Significance:** 3
**Originality:** 3
**Rating:** 4
**Confidence:** 4

**Summary:**

This paper introduces Puzzles, a data augmentation strategy designed to enhance the performance of multi-view 3D reconstruction by synthesizing video-depth sequences from single images or short video clips. Puzzles simulates realistic scene geometry and diverse camera trajectories through targeted patch-based transformations, significantly enriching the diversity of training data. Furthermore, Puzzles is a plug-and-play framework that can be seamlessly integrated into existing 3D reconstruction pipelines without requiring modifications to network architectures. Experimental results demonstrate its effectiveness, showing that models trained on just 10% of the original data augmented with Puzzles can achieve or even surpass the accuracy of full-data baselines. This data-centric approach provides a scalable solution for improving existing 3D reconstruction models.

**Questions:**

See weakness

**Ethical Concerns:**

["NO or VERY MINOR ethics concerns only"]

**Final Justification:**

The proposed data augmentation method, Puzzles, effectively improves the performance of *3R methods with even 10% data. Puzzles is flexible and can be easily applied to different models. Results in the rebuttal addressed most of my concerns. The analysis of camera pose distribution during training also shows a deep understanding of the proposed method making it more insightful.

**Limitations:**

yes

**Quality:**

2

**Strengths And Weaknesses:**

Strengths

1. This paper is well-written and easy to follow.

2. The results of the proposed method are promising. It effectively scales the original dataset during training, enabling more efficient use of existing data.

3. The proposed data augmentation method is flexible and can be easily applied to different models.

Weaknesses

1. When using monocular depth maps from MoGe, how is depth consistency ensured between video frames or keyframes? Inconsistent depth could result in poor supervision labels.

2. I am concerned about whether the proposed method can be applied to videos with lower resolutions, e.g., 512x384. The resized patches may be significantly noisier, which could harm training performance.

3. Keyframe selection is commonly used in video-based tasks. It is unclear whether the proposed keyframe selection method performs better than those used in existing approaches, e.g., NICE-SLAM.

4. How efficient is the proposed method for data augmentation? It would be helpful to include an analysis of the time consumption for each step.

5. I noticed that the performance of the selected baselines is much worse than that of recent SOTA models such as VGGT. It would be interesting to explore whether the proposed method is effective when applied to more recent SOTA models.

6. More patch sampling strategies should be discussed and compared in the ablation study. For example, uniformly sampling patches with a fixed overlap ratio or randomly sampling patches, even without overlapping, could be evaluated.

---

> ### Author Rebuttal · Authors · 2025-07-30
>
> ## Weakness and Questions
> #### **Q1**. When using monocular depth maps from MoGe, how is depth consistency ensured between video frames or keyframes? Inconsistent depth could result in poor supervision labels.
>
> * **Answer**:
> In the Image-to-Clip experiment, we test two variants (as shown in Fig. 7C): one uses GT depth, and the other uses predicted depth from MoGe to extract geometry from a single image, which is then used to generate clips. In the Clips-to-Clips setting, to ensure depth consistency, we only use GT depth for augmentation and do not include predicted depth.
>
> ---
>
> #### **Q2**. I am concerned about whether the proposed method can be applied to videos with lower resolutions, e.g., 512x384. The resized patches may be significantly noisier, which could harm training performance.
>
> * **Answer**:
> Yes, applying Puzzles to low-resolution images (e.g., 512×384) may degrade patch quality, which could impact training performance. In our experiments, we use commonly available resolutions such as 1280×720 and 720×1080, which are sufficient to generate high-quality patches and clips. Examples of the resulting outputs can be seen in Fig. 3, demonstrating the visual fidelity preserved at these resolutions.
>
> ---
>
> #### **Q3**. Keyframe selection is commonly used in video-based tasks. It is unclear whether the proposed keyframe selection method performs better than those used in existing approaches, e.g., NICE-SLAM.
>
> * **Answer**:
> Thank you for the suggestion. We will include a comparison and discussion of SLAM-based keyframe selection methods (e.g., NICE-SLAM) in the main paper. While both approaches involve keyframe selection, their objectives and mechanisms differ fundamentally. In SLAM systems like NICE-SLAM, keyframes are selected to guide online optimization—typically focusing on frames that contribute to pose and geometry refinement through reprojection. In contrast, our method operates on pre-calibrated training data, selecting representative frames based on inter-frame similarity to remove redundancy and ensure viewpoint diversity during data augmentation. Due to these fundamental differences, it is difficult to integrate the two approaches for a fair comparison.
>
> ---
>
> #### **Q4**. How efficient is the proposed method for data augmentation? It would be helpful to include an analysis of the time consumption for each step.
> * **Answer**:
> Using Spann3R as an example, we follow the official training setup and train for 120 epochs on the same dataset. On 8×H100 GPUs, training takes 40 hours with Puzzles and 36 hours without. While integrating Puzzles requires longer training time, the additional 10% is a reasonable and acceptable cost given the performance gains it brings.
>
> ---
>
> #### **Q5**. I noticed that the performance of the selected baselines is much worse than that of recent SOTA models such as VGGT. It would be interesting to explore whether the proposed method is effective when applied to more recent SOTA models.
> | Method            | ACC ↓          | Comp ↓         |
> |-------------------|----------------|----------------|
> | VGGT w/o. Puzzles | 0.0178         | 0.0118         |
> | VGGT w/. Puzzles  | 0.0148 (+16.8%)| 0.0101 (+14.4%)|
> Quantitative comparison on 7Scenes.
> * **Answer**:
> VGGT was released in May, around the same time as the NeurIPS deadline (May 15), making it a concurrent work. Due to time and data constraints, we fine-tuned VGGT using the settings from our paper to compare its performance with and without Puzzles augmentation. As shown in the table, on the 7Scenes dataset, Puzzles brings notable improvements—16.8% in ACC and 14.4% in Comp. Like the methods in our paper, Puzzles is compatible with any existing feed-forward 3D reconstruction framework, requiring no changes to the network or training strategy.
>
> ---
>
> #### **Q6**. More patch sampling strategies should be discussed and compared in the ablation study. For example, uniformly sampling patches with a fixed overlap ratio or randomly sampling patches, even without overlapping, could be evaluated.
> | Setting (Baseline: Spann3R)               | ACC ↓  | Comp ↓  |
> |-------------------------------------------|--------|---------|
> | w/o. Puzzles                              | 0.0388 | 0.0253  |
> | w/. Puzzles – Uniformly sampling          | 0.0378 | 0.0248  |
> | w/. Puzzles – Randomly sampling (Default) | 0.0330 | 0.0224  |
> |
> * **Answer**:
> We compared different patch sampling strategies in Puzzles. Using a uniform patch sampling method—where each image is evenly sampled with fixed-size patches and overlap—we found it still outperforms the no-Puzzles baseline. However, our proposed random patch sampling strategy achieves the best results. By generating patches of varying sizes, it better simulates diverse camera motions and captures more scene details, leading to improved model performance.

---

> > ### Comment · Reviewer_SXr5 · 2025-08-05
> >
> > Thanks for the detailed rebuttal from the authors. Most of my concerns have been addressed. I have one additional question: Is the improvement in reconstruction performance also partly attributed to the increased diversity of camera poses? This is because the proposed data augmentation approach effectively increases the diversity of camera poses during training. If possible, I would appreciate it if the authors could provide analyses of the following two aspects regarding camera poses (based on Spann3R is sufficient):
> > 1. A quantitative comparison of camera pose accuracy, with and without using Puzzles.
> > 2. The difference in the distribution of camera poses when training with and without Puzzles. Since figures cannot be provided, the authors could instead report statistics such as the mean and variance of rotation vectors and translation vectors along each axis.

---

> ### Author Response · Authors · 2025-08-06
> **Further Comparison: With vs. Without Puzzles**
>
> * **Q1** A quantitative comparison of camera pose accuracy, with and without using Puzzles.
>
> | Method |  ATE-RMSE(cm)  ↓  (7 Scenes) |
> | ------------- | ----------- |
> | Spann3R w/o. Puzzles | 12.84 |
> | Spann3R w/. Puzzles |  10.32  |
>
> Due to time constraints, we could not present the complete set of camera-pose accuracy results in the main paper. We will include more results about camera pose estimation in the revision. The tables shown here are from an interim validation and already demonstrate that, in addition to better reconstruction quality, Spann3R with Puzzles yields a significant improvement in camera-pose estimation.
>
> ---
>
> * **Q2** The difference in the distribution of camera poses when training with and without Puzzles.
>
> We analyse the difference in camera pose distribution  across the dataset used in the main paper.
>
> **Table 1. Camera-pose means (μ)**
>
> | Statistic      |  |roll (°) | pitch (°) | yaw (°) |
> | -------------- |  ------------ | ------------ | ------------- | ----------- |
> | **Rotation μ** | |−89.6    | 0.04   | −4.7    |
>
> | Statistic         |  | t_x (m) | t_y (m) | t_z (m) |
> | ----------------- | ----------------- | ----------------- | ----------------- | ----------------- |
> | **Translation μ**|  | −1.58         | −1.39         | 2.45          |
>
> Puzzles follow zero-mean symmetric augmentation leading to no obvious changes in mean.
>
> **Table 2. Camera pose standard deviation (σ) before *vs.* after augmentation**
> | Axis                      | σ_orig | σ_new | Increase    |
> | ------------------------- | ---------------- | --------------- | ----------- |
> | Rotation X (roll)         | 15.9 °           | 54.3 °          | **+242 %**  |
> | Rotation Y (pitch)        | 5.5 °            | 52.2 °          | **+853 %**  |
> | Rotation Z (yaw)          | 93.7 °           | 107.2 °         | **+14 %**   |
> | Translation t_x | 31.1 m           | 35.6 m          | **+14.4 %** |
> | Translation t_y | 30.9 m           | 35.4 m          | **+14.6 %** |
> | Translation t_z | 10.2 m           | 20.1 m          | **+96.5 %** |
>
> The puzzles augmentation adds obvious variance changes to every axis. The axes in original dataset were already broad (yaw, t_x, t_y) with only small percentage bump. Axes that were narrow (pitch, roll, t_z) balloon dramatically. After augmentation, most rotation axes now sit in the 50 ° – 110 ° range, and translations cluster around 20 m – 35 m. The augmented pose distribution is now more balanced, exposing the model to a wider range of viewpoints. The larger σ values quantify how strongly Puzzles expands this space, leading to better generalisation to rare poses, greater robustness to atypical camera motions, and ultimately higher model performance.

---

> ### Comment · Reviewer_SXr5 · 2025-08-06
>
> Thanks to the authors for their efforts. I have no further questions. The analysis of camera pose distribution during training and the comparison of camera pose predictions on 7 scenes demonstrate a deep insight into the proposed method. I recommend that the authors add these results to the revision. I will improve my score accordingly.

---

### Official Review · Reviewer_7DC6 · 2025-07-03

**Clarity:** 3
**Significance:** 3
**Originality:** 3
**Rating:** 5
**Confidence:** 4

**Summary:**

This paper introduces a new data augmentation approach for DUSt3R-based multiview 3D reconstruction methods. Given a single view or short clips of a static scene with RGBD data, this augmentation strategy samples new views from the data via patched cropping and camera view pose augmentation to obtain a series of new views with different intrinsics or extrinsics. As 3R-based methods predict point maps aligned to the first frame coordinates, the point maps sampled in new views are naturally aligned and obtained from the original frame.

This data augmentation strategy partially addresses the constraints faced by 3R-based methods that require large-scale data and diverse viewpoints. It efficiently expands limited training data to much larger scales while providing more diverse camera intrinsics and extrinsics. The method achieves competitive performance compared to the same approach trained on only 10% of the annotated data.

The paper describes the data augmentation for image-to-clips data augmentation strategies with 2 options: fixed extrinsics augment intrinsics, and fixed intrinsics augment extrinsics. All via simple homogeneous coordinates augmentation, transformation and optimizations via linear transformations and optimizations with reprojection errors given GT 3D scene points.

As for clips-to-clips augmentations, an overlapping threshold filtering algorithm is applied to select less redundant keyframes. The same augmentation strategies as image-to-clips are then followed for the keyframes.

Experiments demonstrate that the augmentation pipeline consistently improves the performance of Spann3R, Fast3R, and SLAM3R in scene and object multiview reconstruction tasks without requiring any network architecture modifications. Furthermore, this augmentation approach achieves competitive performance compared to the original models trained on the full dataset while using only a fraction of the training data. These results suggest that the augmentation strategy can effectively address data-limited training scenarios.

**Questions:**

1. Is there a statistical analysis of the occlusion areas after applying the augmentation to quantify the extent of these artifacts?
2. How does this augmentation approach compare to traditional computer vision augmentation techniques (cropping, rotation, resizing, color jittering, etc.)?
3. How do the training time and convergence rate compare between experiments with and without data augmentation? Are the experiments conducted with the same number of epochs?

**Ethical Concerns:**

["NO or VERY MINOR ethics concerns only"]

**Final Justification:**

The authors have addressed my concerns regarding image occlusion introduced by the data augmentation pipeline and its impact on the reconstruction task. They also provided helpful comparisons to other augmentation strategies and clarified details related to baseline comparisons. I believe the proposed simple yet effective improvements in data augmentation are important and could play a significant role in training data-centric multiview reconstruction models, especially when annotations are costly. Therefore, it is an accept for me.

**Limitations:**

Yes

**Quality:**

3

**Strengths And Weaknesses:**

### Strengths

1. The data augmentation pipeline is simple, efficient yet effective.
2. Addresses the data scarcity problem by expanding limited training data to much larger scales without requiring additional annotations.
3. The approach is data-centric imporves perfromance on prior work without requiring any modifications to the underlying network architecture.
4. Achieves competitive results compared to models trained on full datasets while using only a fraction of the training data (10% mentioned).

### Weaknesses

1. New views with different intrinsics and extrinsics will introduce artifacts in invisible areas, but may have less effect on the performance of reconstruction tasks.
2. No comparison of this data augmentation to experiments with no data augmentation at all, more traditional data augmentation techniques (cropping, jittering, image rotation, etc.), and this data augmentation to provide a more comprehensive understanding of how important the perspective projection augmentation is for the reconstruction task.
3. It is unclear whether the model performance comparison with and without data augmentation uses the same number of training epochs, and the impact of data augmentation on training efficiency and convergence rate.

---

> ### Author Rebuttal · Authors · 2025-07-30
>
> ## Weakness and Questions
> #### **Q1**. New views with different intrinsics and extrinsics will introduce artifacts in invisible areas, but may have less effect on the performance of reconstruction tasks.
> | Setting                         | ACC ↓     | Comp ↓    | Training Time (Hours) |
> |---------------------------------|-----------|-----------|------------------------|
> | Spann3R (w/o. Inpainting - Default) | 0.0330    | 0.0224    | 40                     |
> | Spann3R (w/. Inpainting)        | 0.0332 (−0.6%) | 0.0221 (+1.3%) | 53 (**+32.5%**)            |
> | Δ Change                        | +0.0002   | −0.0003   | **+13 h**                  |
> |
> * **Answer**:
> During the method design stage, we observed that camera augmentation can introduce occlusion-induced holes in the projected views, especially in newly visible areas. To address this, we experimented with inpainting to fill these regions. However, our results showed no significant performance improvement, while training time increased by 32.5%. This suggests that such artifacts have limited impact on reconstruction performance, and the cost of mitigating them outweighs the benefits.
>
> ---
>
> #### **Q2**. No comparison of this data augmentation to experiments with no data augmentation at all, more traditional data augmentation techniques (cropping, jittering, image rotation, etc.), and this data augmentation to provide a more comprehensive understanding of how important the perspective projection augmentation is for the reconstruction task.
> | Setting                        | ACC ↓  | Comp ↓  |
> |--------------------------------|--------|---------|
> | No augmentation                | 0.0395 | 0.0261  |
> | Cropping + Jittering + Flipping| 0.0388 | 0.0253  |
> | Puzzles                        | **0.0330** | **0.0224**  |
> |
> * **Answer**:
> We compared the performance of Spann3R on the 7Scenes dataset under three settings: (1) no augmentation, (2) traditional augmentation (cropping + jittering + flipping), and (3) our proposed Puzzles. In the official implementation, traditional augmentation includes cropping, jittering, and flipping, which enhance appearance diversity at the image level. In contrast, Puzzles augments data from a spatial perspective by simulating camera motion and introducing diverse viewpoints.
> Our results show that Puzzles leads to more effective generalization, outperforming both the no-augmentation and traditional augmentation settings.
>
> ---
>
> #### **Q3**.It is unclear whether the model performance comparison with and without data augmentation uses the same number of training epochs, and the impact of data augmentation on training efficiency and convergence rate.
> * **Answer**:
> Using Spann3R as an example, we follow the official training setup and train for 120 epochs on the same dataset. On 8×H100 GPUs, training takes 40 hours with Puzzles and 36 hours without. We observe slower convergence when using Puzzles, likely due to increased data diversity and complexity it introduces. While longer training could improve performance, we use the same settings for fair comparison.

---

> > ### Comment · Reviewer_7DC6 · 2025-08-06
> >
> > Thank you authors for the rebuttal. My concerns regarding the image occlusion introduced by the data augmentation pipeline and its impact on the reconstruction task have been addressed. It appears that inpainting in the occluded regions does not help. The comparisons provided between no augmentation, common image augmentation strategies, and the proposed method demonstrate that the proposed approach offers improvements. My third concern, regarding the fairness and effectiveness of the baseline comparisons, has also been clarified. Overall, I find the proposed augmentation approach to be a simple yet effective, particularly valuable for training data-centric multiview reconstruction models when annotations are expensive. Based on the clarifications in the rebuttal, my review remains as acceptance.

---

### Official Review · Reviewer_2KDP · 2025-07-03

**Clarity:** 3
**Significance:** 3
**Originality:** 3
**Rating:** 4
**Confidence:** 4

**Summary:**

This paper introduces a data augmentation framework for 3D reconstruction that generates high-quality multi-view video-depth sequences from a single image or short video clip. It includes two components called Image-to-Clips and Clips-to-Clips, which enhance local and global view diversity, respectively. Experiments show that it can improve reconstruction accuracy even when models are trained on only 10 percent of the original dataset. Additionally, the method is plug and play and does not require changes to existing network architectures.

**Questions:**

Does the use of PnP on small, local depth patches introduce noisy training data due to inaccurate pose estimates? Since image features within a patch are often similar, pose estimation can become unstable. Does such noise propagate into the final 3D reconstruction or affect model performance?

**Ethical Concerns:**

["NO or VERY MINOR ethics concerns only"]

**Final Justification:**

After reading the rebuttal and discussions, I find that most of my concerns have been adequately addressed. I will therefore keep my original score.

**Limitations:**

Yes

**Quality:**

2

**Strengths And Weaknesses:**

Strengths

1) The paper introduces a novel two-part data augmentation pipeline that directly addresses the data scarcity problem in 3D reconstruction. The idea is simple yet well-motivated, and the subsequent methodology is theoretically sound and technically solid.

2) It requires no changes to network architectures and can be seamlessly integrated into various state-of-the-art 3D reconstruction pipelines such as Spann3R, SLAM3R, and Fast3R.

3) The paper evaluates the method across multiple benchmarks (7Scenes, NRGBD, DTU) and models, under various data regimes, demonstrating consistent and robust improvements.

Weaknesses

1) The augmentation pipeline consists of several components,such as different patch calibration strategies, camera rotation augmentation, and view validation check, but their individual contributions are not isolated. An ablation study would clarify which components are most impactful and whether all are necessary.

2) While the paper highlights that Puzzles enables models trained on only 10% of the data to match or exceed full-data performance, it does not include a baseline where the same 10% is used without Puzzles.

---

> ### Author Rebuttal · Authors · 2025-07-30
>
> ## Weakness
> ####  **Q1**. An ablation study clarify which component are more important
>
> | I2C w/o. aug. | I2C w/. aug. | C2C | Performance ACC ↓ (7 Scenes) |
> |:-------------:|:------------:|:---:|:-----------------------------:|
> | ✓             |              |     | 0.1278                        |
> | ✓             | ✓            |     | 0.0916                        |
> | ✓             | ✓            | ✓   | **0.0330**                        |
> |
> * **Answer:**
> We further explore the ablation study of Figure 7.A to illustrate the impact of each component. The experiments show that Clips-to-Clips (C2C) bring the most significant performance gains, while Image-to-Clips (I2C) with augmentation (w/. aug.) provides a smaller but still positive improvement. In scene-level reconstruction, I2C with augmentation helps simulate camera motion, enabling the network to capture local details. In contrast, C2C improves robustness and supports large-scale reconstruction by capturing global context.
>
> ---
>
> #### **Q2**.  Experiment include 10% data without Puzzles is required.
> | Method               | Data      | ACC ↓         | Comp ↓         |
> |----------------------|-----------|----------------|----------------|
> | Spann3R w/o. Puzzles | 1/10      | 0.0542         | 0.0325         |
> | Spann3R w/. Puzzles  | 1/10      | 0.0389 (+28.2%)| 0.0248 (+23.7%)|
> |
> | Spann3R w/o. Puzzles | full      | 0.0388         | 0.0253         |
> | Spann3R w/. Puzzles  | full      | 0.0330 (+14.9%)| 0.0224 (+11.4%)|
> Quantitative comparison on 7Scenes.
> * **Answer:**
> We have supplemented the evaluation with an experiment where only 10% of the training  data is used without Puzzles augmentation. The improvement shown in the above table are based on the comparison under the same data size. Compared to the baseline using 10% of the data, applying Puzzles augmentation significantly improves reconstruction quality. Notably, under this limited data regime, the ACC and Comp errors are reduced by 28.2% and 23.7%, respectively. This demonstrates that Puzzles greatly improves data efficiency. While the original setting relies on large-scale data to achieve diversity, Puzzles maximizes the utility of available data by generating more informative and varied training samples.
>
> ---
>
> ## Questions
> ####  **Q1**. Does the inaccurate pose estimation have an impact of the model performance.
> * **Answer:**
> We acknowledge this issue, the calibration process is not perfect and can introduce pose inaccuracies. For augmented patches that rely on calibration, we re-render the image using the updated camera intrinsics and extrinsics to align it with the existing pointmap. This step helps correct mismatches between the image and pointmap caused by imperfect calibration, thereby mitigating the impact of inaccurate pose estimation on model performance.

---

> > ### Comment · Reviewer_2KDP · 2025-08-07
> >
> > The authors have addressed my concern regarding the comparison using 10% of the training data for the baseline model without Puzzles augmentation. I raised this point because I suspected that the improvement might stem from data redundancy, meaning the baseline could still perform well even with reduced data. The additional results provided in the rebuttal showed a performance drop when only 10% of the data was used, which resolved this concern.
> >
> > Regarding the ablation study, I was initially worried about the method's reliance on heuristic designs and wanted to understand the sensitivity of each module. Although the authors did not respond to this directly, I found that the rebuttal to Q2.2 by Reviewer tQex and Q6 by Reviewer SXr5 partially addressed my concerns, particularly on patch calibration strategies and camera rotation augmentation.
> >
> > As for my question about PnP, it was driven by curiosity about the accuracy of the PnP and its potential effect on 3D reconstruction. The re-render checks alleviated this concern to some extent.
> >
> > Overall, I believe the authors proposed a simple yet effective method. What stands out to me is its potential as a general-purpose 3D data augmentation strategy that is architecture-agnostic and task-agnostic. However, due to the limited theoretical contribution and reliance on heuristic components, I consider this a borderline accept. I will therefore keep my original score.

---

### Official Review · Reviewer_tQex · 2025-07-04

**Clarity:** 3
**Significance:** 3
**Originality:** 3
**Rating:** 4
**Confidence:** 4

**Summary:**

The paper proposes a data augmentation scheme that can generate novel video clips of posed RGBD images, given a single RGBD image, or a posed video of such images.
This kind of data generation is motivated by the need of current end-to-end dense 3D resonstruction methods in large amounts of labeled data. Such data is relatively scarce and dedicated augmentation schemes are not available.
The solution is based on sampling a sequence of overlapping patches, within a given input image, which are later augmented in both 2D and 3D to create a synthetic sequence of images that simulate coherent and challenging camera motion in the scene.
The benefits of the method are demonstrated by showing accross the board improvements of the several relevant '3R' schemes after training with the additional generated data.

**Questions:**

1) Please explain in more detail how the augmentation scheme is incorporated at training.
2) Regarding image-to-clip:
    - How are the patches actually selected?
    - In patch calibration:
         a. Why do you allow changing either intrinsics or extrinsics, but not both at the same time?
         b. When using fixed-intrinsics-varying-extrinsics, Doesn't it incur only change in translation? (You say it doesn't create *realistic* rotation dynamics)
         c. Why is RanSAC needed in PnP? What are the outliers?
    - In camera motion augmentation, there are the valid view coverage checks, but it seems that you do no enforce overlap between consecutive frames. For example, in bottom row of Figure 3, samples 2 and 3 seem to have no overlap.
    - You create occluded regions, due to camera motion, which you color white and do provide depth. What is the extent of this and does it have impact on the methods that are trained? Should you perhaps have considered image/depth inpainting?
3) Regarding clip-to-clip: You decide to subsample the clip images and then apply image-to-clip on each. Wouldn't you get more general solutions by treating the whole sequence as an input and sampling a sequence of patches from different frames? (i.e. two overlapping patches are more realistic if not sampled from the same frame, but say from consecutive frames)
4) Table 1: Are your results using clip-to-clip or image-to-clip? (And same question regarding Figure 7 B and C).

**Ethical Concerns:**

["NO or VERY MINOR ethics concerns only"]

**Final Justification:**

I want to thank the authors for their comprehensive rebuttal in response to my review. Most of my concerns were properly addressed — especially those regarding clarifications and technical details. I would suggest updating the manuscript accordingly.

My cautious rating was mainly based on the impression that the solution is rather heuristic and lacks a more direct formulation of what video‑based 3D reconstruction pipelines actually require from their training data. Such an understanding, which is still lacking, could enable more rigorous solutions to this problem.

Nevertheless, I acknowledge that this situation is common to most works in this area. However, even if the general goals are to obtain more data, greater diversity, larger motion, etc., these should be better quantified and compared — before and after the augmentations. An important addition in this direction is the table introduced in response to Reviewer SXr5, which analyzes camera pose distributions.

Given this understanding (and the authors’ efforts to improve their experimentation in this area), as well as the fact that the method is highly practical with clear benefits in usage, I have raised my score and would not object to accepting this paper.

**Limitations:**

Yes.

**Quality:**

3

**Strengths And Weaknesses:**

Strengths:
-------------
1) I find the main bottom-line results very convincing. Especially Table 1, that reports for the combination of 3 different leading methods (Spann3R, Fast3R and SLAM3R) over 3 main datasest (7Scenes, NRGBD and DTU), showing consistent improvements when adopting Puzzles (10%-20% gains in accuracy and completion metrics) and comparable results when using (augmenting) only 10% of the original data.
2) The idea of working on a sequence of crops in order to simulate a video sequence is very appealing. It can be used both to generate a huge number of clips, with great controll on variety and diffuculty of the generated data, in terms of overlap camera pose dynamics and scene coverage.
3) The extension from single image input to using clips is challenging, but the suggested procedure is able to extract longer and more complex clips from data that contains lots of overlap. In some sense, it is able to extract good summarizations of this highly redundant data.

Weaknesses:
-----------------
1) The background motivation is a rather limited and it is not entirely clear what the target function is. I would expect to see a more detailed discussion regarding how the mentioned '3R' methods process data - What the requirements are, and What of those is missing in current datasets.
2) This would clarify the design choices of the augmentation method. Currenly, there is a general notion of the need to produce more data, with very vague requirements like being multi-view and temporally consistent, improving spatial coverage, none which are defined or quantified in any way. Therefore, it is hard to tell how much the specific design choices are optimal. Typically, when proposing a new data-set or even such an augmentation scheme, it is beneficial to present statistics of the data, before and after, to show how the new data is more diverse, complete, balanced, challenging, etc'.
3) Desing choices are very much heuristic. I wonder whether it would be possible to make them in a more general way, trying to optimize over some specific functional that captures the desired properties of the clip.
4) Experimentation - It's not clear to me at which extent the scheme is used in the training. In terms of the amount of generated data, whether it is used instead of along with the original data, whether it is done on the fly or in pre-processing and how it effects the training dynamics (e.g. number of iterations, convergence).

---

> ### Author Rebuttal · Authors · 2025-07-30
>
> ## Weakness:
> #### **Q1**. Lack of 1) background motivation, 2) clear target function and 3) detailed data processing.
> * **Answer**: 1) Data-driven feed-forward 3D reconstruction methods, such as VGGT, have shown great promise, with data quantity and quality of training data being key to their success. Existing feed-forward methods primarily rely on 2D data augmentation techniques such as cropping, clipping, and jittering, without incorporating any 3D-aware augmentation. In contrast, our method enhances viewpoint diversity by explicitly controlling camera motion, leading to a more diverse and balanced data distribution in 3D space. 2-3) The task involves taking multiple images (extracted from a video) as input, without requiring any camera parameters, and outputs a pointmap (point cloud) for each image. All pointmaps are expressed in the coordinate frame of the first image. The training data consists of video clips along with their corresponding pointmap.
>
> ---
>
> ####  **Q2**. Hard to tell how much the specific design choices are optimal, which lacks clear definition and quantitative analysis.
> * **Answer**: Directly evaluating the effectiveness of data augmentation is inherently challenging. Instead, we provide both an analysis of the augmented data distribution and its impact on  feature representations. On the ScanNet++ dataset, we observe that the original camera pose distribution is dense along the Yaw and Pitch axes but sparse along the Roll axis, due to the nature of indoor scanning trajectories. After applying Puzzles, the distribution becomes more balanced across all three axes. At the feature level, we extract representations using Spann3R models trained with and without Puzzles. Features obtained with Puzzles exhibit a more uniform distribution, indicating better coverage of the representation space. While we are unable to include figures in the rebuttal due to space constraints, we will provide detailed visualizations and expanded discussion in the main paper.
>
> ---
>
> ####  **Q3**. Design choices are heuristic, need a general way.
> * **Answer**: While our method involves heuristic components, similar to many existing data augmentation techniques, it is designed to be general and broadly applicable. Puzzles functions as a **plug-and-play** module that can be integrated with any feed-forward 3D reconstruction model, whether it operates on image pair or sequences. It requires **no modification to the network architecture** and can be seamlessly incorporated at the data loading stage to enhance viewpoint diversity.
> Furthermore, our method aligns well with current training paradigms. Many reconstruction models adopt **local-to-global training strategies**, where the early warm-up stages involve high data overlap, and later stages reduce overlap to promote broader scene coverage. Puzzles naturally supports this approach: its **Image-to-Clips module allows control over patch overlap**, allowing flexible adaptation of augmentation strength across training phases.
>
> ---
>
> ####  **Q4**. Lack of experimentation details including training scheme, training data, on the fly or pre-processing, training time, iteration and convergence.
> * **Answer**: Using Spann3R as an example, we follow the official training setup with the same 14M training samples (ScanNet-v2, ARKitScenes, Habitat subset, and BlendedMVS). The only change is the application of Puzzles augmentation. Puzzles is a plug-and-play module applied **on-the-fly** through the dataloader, without requiring any pre-processing or modification of the training pipeline. Training was conducted on 8×H100 GPUs for 120 epochs. The total training time was approximately 40 hours with Puzzles, compared to 36 hours without. We observed slightly slower convergence when using Puzzles, which is expected due to the increased diversity and complexity of the training data. While longer training could yield further improvements, we use the same configuration across experiments to ensure fair comparison.
>
> ---
>
> ## Questions
> ####  **Q1**. More details about augmentation
> * **Answer**: Our augmentation method is designed to be easily integrated into any feed-forward 3D reconstruction training pipeline. For each training batch, augmentation is applied at the video clip level. Given a clip with N images and their corresponding point clouds, we first select key representative frames using an overlap-based clip-to-clip matching matrix. Then, we apply image-to-clip augmentation to synthesize new training samples. The augmented data retains the same format as the original—each clip contains of a sequence of images along with their corresponding pointmap. This consistency ensures compatibility with existing architectures and training workflows.
>
> ---
>
> ### 2. Image-to-Clips
> ####  **Q2.1**. How are the patches actually selected?
> * **Answer**: To generate N patches from a single image, the first patch is generated randomly. Each subsequent patch is then created to ensure it overlaps with **at least one of the previously generated patches**, based on IoU calculations. This overlap constraint helps maintain spatial continuity and supports consistency across views. The patch size is determined dynamically, based on the image resolution and the desired number of patches per image.
>
> ---
>
> #### **Q2.2**. In patch calibration: a. Why do you allow changing either intrinsics or extrinsics, but not both at the same time? b. When using fixed-intrinsics-varying-extrinsics, Doesn't it incur only change in translation? (You say it doesn't create realistic rotation dynamics) c. Why is RanSAC needed in PnP? What are the outliers?
> * **Answer**:
> * **a**: Varying both intrinsics and extrinsics simultaneously is theoretically possible, but it causes unnecessary computational overhead. Given that our training data consists only of images and their corresponding pointmap—without access to ground-truth camera parameters—we prioritize simplicity and robustness. A lightweight calibration strategy suffices to generate diverse and effective image–pointmap pairs.
> * **b**: Without augmentation, generating patches only simulates camera translation. As shown in Fig. 2C, all cameras views remain largely forward-facing, limiting rotational diversity. In contrast, our augmentation introduces varying extrinsics, which results in more realistic and diverse camera orientation, as illustrated in Fig. 2D.
> * **c**: The training data includes real-world samples collected via sensors and manually calibrated, which inevitably introduces noise and outliers. Using RANSAC PnP helps mitigate the effect of these errors. It helps filter out erroneous correspondences between 2D image features and 3D points that could otherwise degrade pose estimation.
>
> ---
>
> #### **Q2.3**. No enforce overlap between consecutive frames.
> * **Answer**: Yes, in our augmentation strategy, each newly generated patch is required to overlap with at least one of the previously generated patches, but not necessarily with consecutive frames. This design choice accommodates scenarios with rapid camera motion, where adjacent frames may have little or not overlap.
>
> ---
>
> #### **Q2.4**. The impact of filling occluded regions with white colour.
>
> | Setting                         | ACC ↓     | Comp ↓    | Training Time (Hours) |
> |---------------------------------|-----------|-----------|------------------------|
> | Spann3R (w/o. Inpainting - Default) | 0.0330    | 0.0224    | 40                     |
> | Spann3R (w/. Inpainting)        | 0.0332 (−0.6%) | 0.0221 (+1.3%) | 53 (**+32.5%**)            |
> | Δ Change                        | +0.0002   | −0.0003   | **+13 h**                  |
> |
> * **Answer**:
> We also observed this issue. We applied LAMA [1] to inpaint occluded regions and excluded the inpainted areas from supervision. However, our experiments conducted on 8×H100 GPUs show that inpainting does not provide a significant performance improvement, while increasing training time by 32.5%. Additionally, existing inpainting methods tend to introduce artifacts when filling large missing regions, contaminating the valid data. Therefore, we decided to discard this approach.
>
> Reference:[1] Resolution-robust Large Mask Inpainting with Fourier Convolutions
>
> ---
>
> #### **Q3**. Clips-to-Clips. Render a new sequence on point cloud during training.
> * **Answer**: Pre-processing Hypersim data using this method is time-consuming—it takes about 3 seconds per clip and around 9 hours for the entire dataset. Therefore, re-rendering scenes on the fly during training is currently not feasible. That said, we agree it is a promising direction, especially for large-scale scene reconstruction (as mentioned in the supplementary), where stitching multiple scenes and generating expansive training data could offer significant benefits.
>
> ---
>
> #### **Q4**. Table 1: Are your results using clip-to-clip or image-to-clip? (And same question regarding Figure 7 B and C).
> * **Answer**: Thanks for pointing out. The results in Table 1 and Figure 7 B are based on the clip-to-clip setting, while Figure 7C corresponds to the Image-to-Clip setting.

---

> > ### Comment · Reviewer_tQex · 2025-08-06
> > **Updated decision**
> >
> > I want to thank the authors for their comprehensive rebuttal in response to my review.
> > Most of my concerns were properly addressed — especially those regarding clarifications and technical details. I would suggest updating the manuscript accordingly.
> >
> > My cautious rating was mainly based on the impression that the solution is rather heuristic and lacks a more direct formulation of what video‑based 3D reconstruction pipelines actually require from their training data. Such an understanding, which is still lacking, could enable more rigorous solutions to this problem.
> >
> > Nevertheless, I acknowledge that this situation is common to most works in this area. However, even if the general goals are to obtain more data, greater diversity, larger motion, etc., these should be better quantified and compared — before and after the augmentations. An important addition in this direction is the table introduced in response to Reviewer SXr5, which analyzes camera pose distributions.
> >
> > Given this understanding (and the authors’ efforts to improve their experimentation in this area), as well as the fact that the method is highly practical with clear benefits in usage, I have raised my score and would not object to accepting this paper.

---

### Note · Authors · 2025-08-14

We sincerely appreciate the reviewers' thoughtful and constructive engagement. We are encouraged by the overall recognition of our paper's **novelty**, **quality of presentation** and **practical contribution**.

**Puzzles** is a plug-and-play video-depth augmentation, that address data scarcity and skewed viewpoint distributions in end-to-end 3D reconstruction.  By synthesising view-consistent RGB-D clips with controllable camera motion at the dataloader (no model changes), Puzzles improves model performance, especially in low-data regimes, and  transfers across architecture / datasets.

This was an excellent submission experience—the reviewers’ deep expertise provided invaluable guidance. In particular, the points from **[tQex, 7DC6]** on training time and implementation details, **[2KDP]** on the 10% data baseline without Puzzles, **[7DC6]** on comparisons with conventional augmentations, and **[SXr5]** on post-Puzzles data distribution and sampling strategies prompted important, meaningful experiments. In the rebuttal, we incorporated these comparisons and analyses, addressing most reviewer concerns; the suggestions and ensuing discussion will also substantially strengthen our revision. Thank you again for the insightful feedback.

Finally, we believe Puzzles will push the state of feed-forward 3D reconstruction and inspire further work on 3D-augmentation across downstream applications.

---

### Decision · Program_Chairs · 2025-09-17

**Decision:**

Accept (poster)

**Comment:**

This paper received unanimously positive reviews after the rebuttal (1 accept and 3 borderline accepts).

All of the reviewers recognize the core contribution of this paper, the proposed data augmentation scheme, being simple, well-motivated, and effective. It can serve as a plug-play module into Dust3R variants without modifications of the model architecture and training objectives.  The strong empirical results are also recognized by the reviewers.

There were concerns raised after the initial reviewing stage. Notably, Reviewer tQex questioned the heuristic nature of the design choices and requested more details on how the augmentation scheme is used in the training. Reviewer 2KDP pointed out the need for a baseline using 10% of the data without Puzzles and requested a more detailed ablation study. Reviewer 7DC6 asked for a comparison against traditional data augmentation techniques, while Reviewer SXr5 raised questions about the method's efficiency and applicability to newer SOTA models. The authors addressed these comments in the rebuttal, including providing new experimental results of ablation studies and integrating the proposed data augmentation scheme into VGGT.

Although minor concerns still remain, mainly about the heuristic nature of the proposed data augmentation scheme and lack of systematic understanding of its effectiveness, the AC found that the proposed approach is solid, tackles an important problem, and may inspire future research in this direction.

The authors are highly recommended to incorporate the results reported in the rebuttal into the final revision.